# Analysis of Perception, Reasons, and Motivations for COVID-19 Vaccination in People with Diabetes across Sub-Saharan Africa: A Mixed-Method Approach

**DOI:** 10.3390/ijerph19137875

**Published:** 2022-06-27

**Authors:** Uchechukwu Levi Osuagwu, Raymond Langsi, Godwin Ovenseri-Ogbomo, Khathutshelo Percy Mashige, Emmanuel Kwasi Abu, Esther Awazzi Envuladu, Piwuna Christopher Goson, Bernadine Nsa Ekpenyong, Richard Oloruntoba, Chundung Asabe Miner, Deborah Donald Charwe, Chikasirimobi Goodhope Timothy, Tanko Ishaya, Onyekachukwu Mary-Anne Amiebenomo, David Lim, Kingsley E. Agho

**Affiliations:** 1Translational Health Research Institute (THRI), School of Medicine, Western Sydney University, Sydney, NSW 2560, Australia; david.lim@westernsydney.edu.au (D.L.); k.agho@westernsydney.edu.au (K.E.A.); 2African Vision Research Institute (AVRI), Discipline of Optometry, University of KwaZulu-Natal, Westville Campus, Durban 3629, South Africa; mashigek@ukzn.ac.za; 3Health Division, University of Bamenda, Bambili P.O. Box 39, Bamenda, Cameroon; raylangsi@yahoo.com; 4Department of Optometry, Centre for Health Sciences, University of the Highlands and Islands, Inverness IV2 3JH, UK; godwin.ovenseri-ogbomo@uhi.ac.uk; 5Department of Optometry and Vision Science, School of Allied Health Sciences, University of Cape Coast, Cape Coast 03321, Ghana; eabu@ucc.edu.gh; 6Department of Community Medicine, College of Health Sciences, University of Jos, Jos 930001, Nigeria; envuladue@unijos.edu.ng (E.A.E.); minerc@unijos.edu.ng (C.A.M.); 7Department of Psychiatry, College of Health Sciences, University of Jos, Jos 930001, Nigeria; piwunag@unijos.edu.ng; 8Department of Public Health, Faculty of Allied Medical Sciences, College of Medical Sciences, University of Calabar, Calabar 540271, Nigeria; bekpenyong@unical.edu.ng; 9School of Management and Marketing, Curtin Business School Bentley, Perth, WA 6102, Australia; richard.oloruntoba@curtin.edu.au; 10Tanzania Food and Nutrition Center, Dar-es-Salaam P.O. Box 977, Tanzania; deborah.charwe@tfnc.go.tz; 11Department of Optometry and Vision Sciences, Mzuzu University, Mzuzu 105200, Malawi; chikasirimobi.t@mzuni.ac.mw; 12Department of Computer Science, University of Jos, Jos 930001, Nigeria; ishayat@unijos.edu.ng; 13Department of Optometry, University of Benin, Benin 300213, Nigeria; maryanne.amiebenomo@uniben.edu; 14School of Health Sciences, Western Sydney University, Sydney, NSW 2560, Australia

**Keywords:** diabetes, survey, Sub-Saharan Africa, coronavirus, vaccine, hesitancy, refusal, qualitative, lockdown

## Abstract

Diabetes mellitus (DM) is associated with severe COVID-19 infection and complications. This study assesses COVID-19 vaccine acceptance and hesitancy in people with DM, and explores the reasons for not being vaccinated. This was a web-based cross-sectional survey using a mixed-method approach conducted in March–May 2021, corresponding to most Sub-Saharan African (SSA) countries’ early vaccine rollout period. Participants were those aged ≥18 years with self-reported DM in 11 Sub-Saharan African (SSA) countries. Responses to comments on the reasons for vaccine hesitancy and facilitators for vaccine uptake were analyzed. Of the 73 participants with DM, 65.8% were males, older than 35 years (86.3%), had postsecondary education (90%), and a significant proportion were from South Africa (39.7%), Nigeria (28.8%) and Ghana (13.7%). At the time of this study, 64.4% experienced COVID-19 symptoms, 46.6% were tested for COVID-19, of which 19.2% tested positive. Few participants (6.8%) had received a COVID-19 vaccination, 65.8% were willing to take the vaccine when it becomes available in their country, while 26.0% either refused or remained hesitant towards taking the vaccine. The main identified reasons for not taking the vaccine were: advice from religious leaders; concerns about the safety, effects, and efficacy of the vaccines; mistrust of the pharmaceutical companies producing the vaccines and the process of production; the conspiracy theories around the vaccines; and the personal belief of the participants regarding vaccination. However, participants stated they would take the vaccine if they were more educated about it, received positive feedback from those vaccinated, were rewarded for taking the vaccine, or if vaccination became a condition for travel and employment. In conclusion, this study shows that the uptake of the COVID-19 vaccine was very low in this high-risk group. Efforts to increase the uptake of COVID-19 vaccines among people with diabetes are imperative, such as the provision of education and relevant information.

## 1. Introduction

Diabetes mellitus (DM) has reached epidemic proportions, globally affecting approximately 463 million people [1]. It is a leading cause of avoidable hospitalizations, amputations, cardiovascular events, renal failure, fetal malformations, and blindness [2]. Three-quarters of those with DM live in low- to middle-income countries (LMIC), and this is projected to increase [3], mostly due to increasing urbanization, demographic, and nutritional changes in the region [4,5,6]. Studies found that Sub-Saharan Africa (SSA) is particularly affected by obesity and diabetes [7]. This risk represents a substantial challenge for the overburdened healthcare systems in the region faced by COVID-19 challenges [8]. Understanding the challenges of this high-risk group is crucial to COVID-19 recovery and lessons learnt in informing preparation for future pandemics.

Prior to the development and rollout of COVID-19 vaccines, stringent lockdowns and other public health safety measures were the predominant methods used to curb the spread of the SARS-CoV-2 virus [9]. Some of these measures, such as stay-at-home policies and mandatory quarantine, promote a sedentary lifestyle, leading to an increase in obesity. This predisposes individuals to a greater risk of poor glycemic control due to physical inactivity. Following the global development and rollout of COVID-19 vaccines, most governments, especially in developed countries, rapidly vaccinated their populations as a public health preventative measure to contain the spread of the virus [10]. However, access and distribution of COVID-19 vaccines remain an issue in most developing countries, particularly in low-income African countries [10], despite recent global efforts to render the vaccines affordable and available to these countries.

The development of the COVID-19 vaccines may have come as a welcome relief for people living with DM in terms of the opportunity to resume outdoor exercises and activities, and in terms of their higher risk of infection and severe complications from being infected with the virus [11,12]. People with DM have poorer health outcomes, such as higher hospitalization and mortality rates from COVID-19 infection compared with their counterparts without the disease [11,12]. As a result, most governments shifted their focus from previously prioritizing healthcare workers to receive the vaccine to including those with chronic illnesses, including DM [13]. In some SSA countries such as Cameroon, priority for vaccination was given to people with Types 1 and 2 DM who were on two or more medications [14]. 

The breakthrough in COVID-19 vaccine discovery, manufacture, and availability was accompanied by issues of myths and misinformation, followed by resistance and hesitancy [15]. There is a paucity of information on COVID-19 vaccine hesitancy among people with DM in SSA countries. Higher prevalence of vaccine hesitancy in Italy (14.2%) [16] and Saudi (29.0%) [17] was reported for individuals with DM. In SSA, people with DM are not left out of these controversies, with reports of mistrust for pharmaceutical companies and concerns about the safety of the COVID-19 vaccines featuring prominently in previous studies conducted elsewhere [16,17]. However, similar concerns with vaccine safety and side effects, the lack of trust in pharmaceutical industries and vaccination trials, and misinformation or conflicting information from the media have been expressed among the general population in SSA countries [18,19]. 

Considering that almost every person needs to receive the COVID-19 vaccine to achieve herd immunity [20], and individuals with DM are more severely affected by COVID-19 disease [21,22], this survey was conducted to assess COVID-19 vaccine acceptance and hesitancy or unwillingness among people with DM in SSA. It also assesses the barriers and beliefs that affect their willingness to receive the COVID-19 vaccine. The findings help in narrowing the current knowledge gap and improving health outcomes for people living with DM in SSA countries.

## 2. Materials and Methods

### 2.1. Study Design

This study used a web-based embedded cross-sectional survey using a mixed-method approach to evaluate the study objectives and responses. The self-administered questionnaire had been validated [23] and was adapted with minor modifications to suit this study’s objective. The questionnaire was pretested with 10 participants who were not included in the final study and were not part of the research group. The pilot study was to ensure clarity and understanding, and to determine the duration for completing the questionnaire prior to dissemination. The survey tool was tested for the internal validity of the items, and Cronbach’s alpha coefficient scores ranged from 0.70 and 0.74, indicating satisfactory consistency [23]. The final survey in English was translated into a French version to allow for wider participation from SSA countries. Researchers from the Department of Linguistics at the University of Bamenda, Cameroon translated the survey tool. There was also a backward translation from French to English to ensure that the meaning of the items was retained.

### 2.2. Ethical Approval

Ethical approval for this study was obtained from the Humanities and Social Sciences Research Ethics Committee of the University of KwaZulu-Natal, Durban, South Africa (HSSREC 00002504/2021). The study adhered to the principles of the 1967 Helsinki Declaration (as modified in Fortaleza, 2013) for research involving human subjects. Participation in this survey was voluntary, and informed consent was sought prior to the survey. The confidentiality of the participants was maintained, and all data were kept anonymous. As part of the preamble, participants were instructed not to take part in the survey more than once, and the IP address of the participants also restricted analysis of the data to prevent multiple and repeat participation in the survey.

### 2.3. Participants

Consenting English- and French-speaking participants aged 18 years and above who had been born in any of the 46 SSA countries and self-reported DM at the time of this survey were eligible to participate in this study. In contrast, data for participants with no record of age or those younger than 18 years, who did not state their DM status, or had no DM were excluded.

### 2.4. Data Collection

The survey was distributed online between March and May 2021. A convenient sampling technique was used to include all the participants in the survey, and those who self-reported DM during the study were extracted. An invitation link to the survey created in Survey Monkey was disseminated in English and French (which are the spoken languages in 21 of 26 SSA countries [24]) using social media platforms (Facebook and WhatsApp) and by email through the authors’ networks. 

The survey items included: 11 sociodemographic variables, items on smoking status; past vaccinations for other conditions (hepatitis, influenza, chickenpox, whooping cough, tuberculosis, yellow fever, measles/mumps/rubella (MMR), diphtheria, pertussis, and tetanus (DPT)), the presence of pre-existing conditions, including heart disease, kidney disease, hypertension, diabetes, obesity, asthma, and sickle cell anemia; knowledge of COVID-19 vaccination; COVID-19 test and result; if they had received any COVID-19 vaccination. For those who had not been vaccinated at the time, a follow-up question was asked to gauge their willingness to receive a COVID-19 vaccine when it became available in their country. This allowed for participants to provide comments on their opinions. There were also questions to understand the participants’ sources of information on COVID-19 and their perception of risk for COVID-19 (four items: three utilizing a Likert scale, and the other a ‘yes’, ‘no’, or ‘not sure’ response). 

### 2.5. Qualitative Responses

Two questions were asked to the participants who were hesitant or refused to receive the COVID-19 vaccines, and their responses were qualitatively analyzed. Both questions required the participants to select from ten options, with an additional section for added comments if they chose to. For the first question, ‘which of the following factors contribute to your decision to not accept a COVID-19 vaccine?, the options were: advice from religious leaders, advice from politicians, mistrust for the pharmaceutical company, mistrust of the health system in my country, mistrust in the medical process for developing the vaccine, mistrust for the country where the vaccine was produced, personal beliefs or past historical experiences with vaccines, concerned about safety of the COVID-19 vaccine, not enough information from healthcare providers, and information from the media. 

For the second question, ‘what can be done to encourage you to get the vaccine?’, the options were: “I am more likely to accept the COVID-19 vaccine (1) if financial incentives are given to everybody; (2) if monetary rewards are given to healthcare providers involved in the vaccination; (3) if it is given for free; (4) if there is adequate information regarding the specific vaccine; (5) if I can get more education on the vaccines, their side effects, and how effective they are; (6) if it is a travel condition; (7) if it is an employment condition; (8) if many people start receiving the vaccine; (9) if I get positive feedback from those who have been vaccinated’.

### 2.6. Statistical Analysis

Statistical analysis was conducted using IBM SPSS Statistics for Windows, version 27 (IBM Corp., Armonk, NY, USA). The frequency and percentage of categorical variables are reported. The proportions of participants with DM who were vaccinated against COVID-19 and those who expressed uncertainty towards being vaccinated were determined. The vaccinated group were those who responded in affirmation (Yes) to the question ‘*have you been vaccinated against COVID-19?*’ Similar to a previous study [25], those who responded ‘not sure’ or ‘no’ regarding being vaccinated against the COVID-19 vaccine were asked if they were willing to be vaccinated when the vaccine became available in their home countries. The responses of ‘not sure’ or ‘no’ to the follow-up question were used to derive the ‘hesitant or refused to accept COVID-19 vaccine’ estimation, and the association with demographic variables was determined with Fisher’s exact test due to the small number of persons in the cells.

The selected options and the open-ended comments obtained from the qualitative section of the questionnaire on their reasons for not receiving the vaccination and what would encourage them to become vaccinated were grouped into major topics and analyzed qualitatively. The significant recurrent and silent points are also reported using quotations, and their frequencies are reported descriptively.

## 3. Results

### 3.1. Characteristics of the Study Population

Of the total of 2572 participants in the general survey, only the responses from 73 (2.94%) participants with self-reported DM (of any type) were used in this study. Figure 1 shows the flowchart of the participant selection from the larger study population.

The characteristics of the study sample presented in Table 1 show that the majority (65.8%) were males, more than two-thirds were aged 35 years and older, and most had at least a tertiary education (90% had either a diploma, university, or higher education), the majority were married (72.6%) and employed (79.5%), and few were working in a healthcare sector (31.5%) (Table 1).

Figure 2 presents the participants’ countries of origin, indicating that the participants were mostly from South Africa (39.7%), Nigeria (28.8%) and Ghana (13.7%), while other SSA countries had minimal participation.

There were few smokers (12.3%), and nearly all the participants reported they had been vaccinated for other conditions, mostly yellow fever, tuberculosis, polio, and hepatitis, which are shown in Figure 3.

Table 2 presents the self-reported health conditions of the participants. More than half (63.0%) had other chronic diseases, mostly hypertension (46.6%), and obesity (20.5%), and related cardiovascular diseases coexisting with diabetes.

### 3.2. Information Related to COVID 19 and Vaccination Uptake

Approximately two-thirds (64.4%) of the participants had had a symptom of COVID-19 during the pandemic, about half had been tested for COVID-19 at least once (46.6%), and 41.2% (*n* = 34) had tested positive for COVID-19. Nearly every one of the participants was aware that COVID-19 vaccines had been developed. However, only 5 of the 73 people with DM (6.8%) had already received a COVID-19 vaccine (34.7%), while the rest were either hesitant (65.8%) or refused (26.0%) to be vaccinated. Most participants (93%) believed that COVID-19 was real, and more than half either agreed or strongly agreed that the vaccine could protect or prevent them from contracting COVID-19 infection. Regarding their perception of risk, about two-thirds (64.3%) felt that they were at risk of contracting the virus, and a slightly lower proportion thought they could die from the infection if they contracted the virus (Table 3).

The sources through which the participants obtained information related to COVID-19 are presented in Figure 4. The figure shows that about 88% of the participants obtained information from Internet sources during the pandemic, including via personal search on Google, scientific journals, health websites (WHO, CDC, and Ministry of Health websites). A breakdown of the various internet sources listed by the respondents is presented as a Appendix A. Social media were the second highest source of information used by the participants during the pandemic, followed by TV, while the least-cited source of information was newspapers, which were used by slightly more than half of the participants (52.1%) for COVID-19-related information during the pandemic.

### 3.3. Association between Hesitancy or Refusal towards COVID-19 Vaccine and the Study Variables

Results of Fisher’s exact test revealed significant associations between hesitancy or refusal of vaccine and sex, region of origin, and place of residence (local and diaspora, see Table 4). In addition, a significant proportion of those who had expressed concern about the safety of the vaccine were less likely to receive the COVID-19 vaccine when it became available in their country (*p* < 0.0001) compared with those who had no such concern. Compared with participants who had not been tested for COVID-19 at the time of this study, significantly more people who had performed a COVID-19 test were willing to accept the vaccine. Other variables not shown in Table 4, including previous vaccination history and the presence of other health conditions such as hypertension, did not show significant association with COVID-19 vaccine hesitancy or refusal.

### 3.4. Reasons for COVID-19 Vaccine Hesitancy or Refusal among Participants

The responses of 19 participants who had said they were either not sure or unwilling to receive the COVID-19 vaccine when it became available in their respective countries were categorized into 7 major headings. These are represented in Figure 5, and the participants’ statements are presented in Appendix A. From the figure, two in five people who were unwilling to take the vaccine cited advice from their religious leaders as their main reason, followed by 22% of participants, including three pregnant individuals and one with a compromised immune system, who cited concerns about the safety of the vaccines as the main reason for not receiving them (Appendix A). A participant from an Eastern African country said, ‘vaccines have been used against black people for far too long—Kenya infertility, Tuskegee, etc.’, while another participant from West Africa said, ‘this vaccine is questionable, and its benefits for politicians far outweigh its care to manage this self-limiting bug’. Few participants did not trust pharmaceutical manufacturing companies, the countries where the vaccines were produced, or the manufacturing process of the vaccine, which was reflected in the following participant statements: ‘the vaccines have been developed so quickly’; ‘I don’t trust the research done about it’; ‘there is not enough scientific data on clinical trials for the vaccines’, and ‘I would rather prefer self-protection for prevention purposes than trust the vaccine’. Few others refused the vaccine due to personal beliefs or past historical events, including one person who stated, ‘I have not seen a need for it’. For other participants, different conspiracy theories circulating regarding the COVID-19 vaccine discouraged them from taking the vaccine, with statements such as ‘the vaccine is meant to reduce world population, especially Africans’, ‘It could be a birth control procedure to reduce world population’ (see Appendix A). The lack of trust in their country’s health system was reported by a few participants as the main reason for not accepting the vaccine, ‘my country is making money with COVID-19, and there is no trace of the disease here’, and someone mentioned that refusal to take the vaccine was because of the advice received from some politicians.

### 3.5. Factors That Would Encourage Uptake of Vaccination in People with DM

The following factors were reported to influence the participants’ decision towards receiving COVID-19 vaccination when it became available in their countries. Education about the vaccines (*n* = 19), was top in the list of factors that would encourage uptake of COVID-19 vaccines, and this was reflected by this statement: ‘If I can get more education on the vaccines, their side effects, and how effective they are, I will take it’. More than half of the participants (*n* = 11) said they would consider receiving the vaccine if they get positive feedback from those who already got it ‘I will decide after I hear positive feedback from those who have been vaccinated’. Four participants reported that high uptake in the population (i.e., after many people had already received theirs) and if the vaccination was free would encourage them to receive the vaccine, as reflected in the following statements: ‘I will accept the vaccination only after many people start receiving the vaccine’ and ‘I will accept the vaccination if I don’t have to pay or bribe someone to get vaccinated’, respectively. Others were more likely to accept the COVID-19 vaccine if there was a monetary reward for receiving it (*n* = 2): ‘I will accept the vaccine if monetary rewards are given to health care providers involved in the vaccination’, or if it was an employment or travel requirement (*n* = 3).

## 4. Discussion

The COVID-19 pandemic has caused massive global disruption in diagnosing and treating chronic illnesses such as DM, leading to an increase in morbidity and mortality risks [26]. This can partially be explained by the increased tendency to manifest a cytokine storm characterized by severe COVID-19 due to disruptions in the cytokine/chemokine pathway [27]. This cross-sectional survey assessed vaccine hesitancy among adults with DM in SSA. The uptake of COVID-19 vaccines was very low among the respondents, even though most of them had had previous vaccinations for other conditions and had pre-existing comorbidities, mostly hypertension, which increases their risk of severe complications. Male participants, those from Central and Southern African countries, people who lived locally, and those who had been tested for COVID-19 were significantly more willing to be vaccinated when the vaccines were available in their country compared with their counterparts in this study. While advice from religious leaders, concern about the safety of the vaccine, and mistrust in pharmaceutical companies were the main reasons behind vaccine hesitancy, participants reported that education about the side effects and efficacy of the vaccines, and receiving positive feedback from those vaccinated would encourage them to become vaccinated. Participants reported that Internet sources, including Ministry of Health websites, were their main source of information, followed by social media platforms.

The World Health Organization (WHO) and Centers for Disease Control and Prevention (CDC) strongly advocate for individuals with DM to be vaccinated against SARS-CoV-2 [28,29]. The very low uptake of COVID-19 vaccines found in this study may be attributed to the very slow rollout of the vaccines due to their unavailability in SSA at the time of this study [10]. It also reflected the overall low rate of COVID-19 vaccination (defined by total vaccines per a population of 100 individuals) in SSA [30], which only rose by 15% between January and February 2022 across the continent due to vaccination campaigns by the SSA countries [31], but remains considerably poor compared with resource-rich developed countries [10]. Evidence demonstrates increased vulnerability to severe COVID-19 illness [32,33] and about three times greater risk of hospitalization in people with DM compared with those without DM [32,34]. The risk of severe infections in our participants could be much higher, with many of them reporting other comorbidities such as hypertension that adversely affect viral clearance, thereby worsening prognosis during a COVID-19 infection [35]. Ensuring that vaccination is promoted among this high-risk group is important due to the poor outcomes of people with DM when admitted due to COVID-19, despite having good glycemic control prior to admission [36].

The present study found that more males than females were willing to take the COVID-19 vaccine. In a review study [37], the authors suggested that the relationship between sex and vaccine acceptance is ambiguous. For instance, whereas some studies reported more COVID-19-hesitant females than males [25,37,38] and vaccine hesitancy generally [39,40], others [17,18] showed that males were more hesitant than females towards vaccination [17]. In a study conducted in Saudi Arabia among people with DM, researchers found that the female sex, the longer duration of diabetes, and having no history of influenza vaccination increased the likelihood of accepting the COVID-19 vaccine after adjusting for potential confounders. The inconclusive findings suggest the need for further research on the impact of sex and to see whether females may respond differently to different health promotion approaches.

Past vaccination history is linked to willingness to be vaccinated [41,42], such as in a UK study where individuals previously vaccinated for the flu were more willing to be vaccinated against COVID-19 compared to their counterparts [41]; in another study, individuals who had previously refused any vaccination were less likely to accept the COVID-19 vaccine [42]. Although nearly all the participants in this study had had previous vaccinations for other conditions, this had no significant effect on the participants’ hesitancy or refusal to receive the COVID-19 vaccine.

Various perceptions are widespread among people with DM, and these may affect their health outcomes. This focus of this study was on individuals with DM in SSA countries to better understand their barriers and enablers to vaccine acceptance. While almost all the participants were aware of COVID-19 vaccines, more than half of them also believed that the vaccines could prevent COVID-19 infection and complications, and nearly all the participants did not believe that the disease was a hoax. The participants reported a high-risk perception of contracting the infection, and about two-thirds felt at risk of being severely infected with the COVID-19 virus, while some of them thought they could die from contracting the infection. These findings agreed with a Dutch study among young adult students that suggested that stronger perceptions of the severity of COVID-19 infection heightened the worries of being infected, but positively influenced the individual’s intentions towards vaccination. This resulted in a higher likelihood of COVID-19 vaccine uptake and vice versa [43]. These perceptions can be traced back to the various sources of information (mainly the Internet and social media platforms) used by the respondents during the early phase of the COVID-19 vaccine rollout in SSA.

The female sex and concern about vaccine safety were key considerations for hesitancy towards COVID-19 vaccination in this study. Facilitators of vaccination uptake included education [44,45,46], which was also reported to improve uptake of the COVID-19 vaccine by the participants. The observation that more education about the vaccine may improve uptake of the vaccine could be attributed to the fact that the majority of participants in this study were educated and may tend to be more curious about the vaccine prior to uptake [47]. A similar finding was reported among healthcare workers, where those with more knowledge about the vaccine mechanism of action and sources reported higher vaccine uptake [48]. A landmark population survey of South Africans in June–July 2021 also found that concerns about the side effects, distrust of government, belief in conspiracy theories, low or no income, and dependence on alternate decision makers were independent predictors of COVID-19 vaccine hesitancy [49]. Therefore, campaigns and sensitization programs geared toward increasing vaccination rates should consider these factors as very essential.

About two in five participants who were hesitant or refused to take the vaccine cited advice from their religious leaders as one of their main reasons for nonvaccinating. This finding is consistent with those of previous reports [50,51] that demonstrated the key role of religious leaders in the uptake of vaccination programs. The attitude of religious leaders towards vaccination varies from full acceptance to clear refusal, and their views are reflected in those of their local congregation members [50]. The beliefs of some religious leaders such as orthodox Protestants who object to vaccination are rooted in religious doctrine; through their authority, they decide how to interpret and apply this doctrine [50]. A study from F Viskupič and DL Wiltse [51] found that, when messages were endorsed by a religious leader, they had a positive and statistically significant effect on people’s interest in becoming vaccinated, whereas those that were endorsed by a political or medical leader had no significant effect. There needs to be a dialog with the religious leaders on how they can help in controlling epidemics by other means than vaccination [51]. While concern about the vaccine’s safety and mistrust in pharmaceutical companies were the other main reasons behind vaccine hesitancy or refusal, participants reported that education about the vaccine, its side effects, and efficacy, and receiving positive feedback from those vaccinated would encourage them to get vaccinated. Participants reported that the ministry website was their main source of information, followed by social media.

### Limitations and Strengths

The study has some limitations that should be considered when interpreting the results or comparing them with those of other studies. First, the small sample limited the generalizability of findings beyond the study, as the survey sample is not representative of all those with DM in the region and of the opinion of the general SSA population. Second, key indicators such as diabetes type and duration, and metabolic measures such as glycated hemoglobin A1c (HbA1c) were not asked because the study was not designed specifically for people with DM. Third, the self-reported DM, COVID-19, and other health conditions were not objectively verified. Fourth, there is the issue of the social desirability factor where the opinion of the participants in surveys might not represent their true opinion on the asked questions. Fifth, the inequality in the vaccine supply across the different SSA regions and the unavailability of vaccines in SSA at the time of this study may have led to an overestimation of the unvaccinated participants. Sixth, as a cross-sectional study, we were unable to determine causation. Seventh, the use of social media platforms and emails for survey distribution may limit the generalizability of the findings because the opinion of rural residents, where Internet penetration remains relatively low [52], and those who do not have access to the Internet may have not been captured. Despite these limitations, this study provided the first evidence from SSA on the reasons for nonvaccinating in a high-risk group of adults who are prone to adverse COVID-19 outcomes. It captured opinions from Francophone and Anglophone countries through the dissemination of the survey tool in both English and French languages. In addition, the analysis of the comments provided qualitative evidence that can be used to inform public health control measures for this and future pandemics.

## 5. Conclusions

The current study investigated COVID-19 vaccine acceptance by exploring many factors, including the effect of history of vaccination, pre-existing conditions, and past diagnosis of COVID-19, and the reasons for vaccine hesitancy and facilitators to COVID-19 vaccine uptake. The findings reveal that the uptake of the COVID-19 vaccine was very low in this high-risk group, and it is imperative that there be efforts to increase the uptake of the vaccine through providing education and information about the vaccine, and some financial incentives. Through these findings, public policies, guidelines, and communication strategies can be formulated to enhance the public’s confidence in the various COVID-19 vaccines. This can lead to an overall increase in vaccine uptake. That many of the participants expressed concern about inadequate information calls for action to provide accurate information through healthcare and community health workers, as this can go a long way towards the success of vaccination efforts in the region. Considering the increased supply of vaccines in the region and the availability of more evidence on the COVID-19 vaccine, similar studies exploring COVID-19 vaccine refusal or hesitancy and the reasons for nonvaccinating are needed, particularly in such high-risk groups, including people with hypertension, to understand their reasons for nonvaccinating and whether this has changed over time for those with DM.

## Figures and Tables

**Figure 1 ijerph-19-07875-f001:**
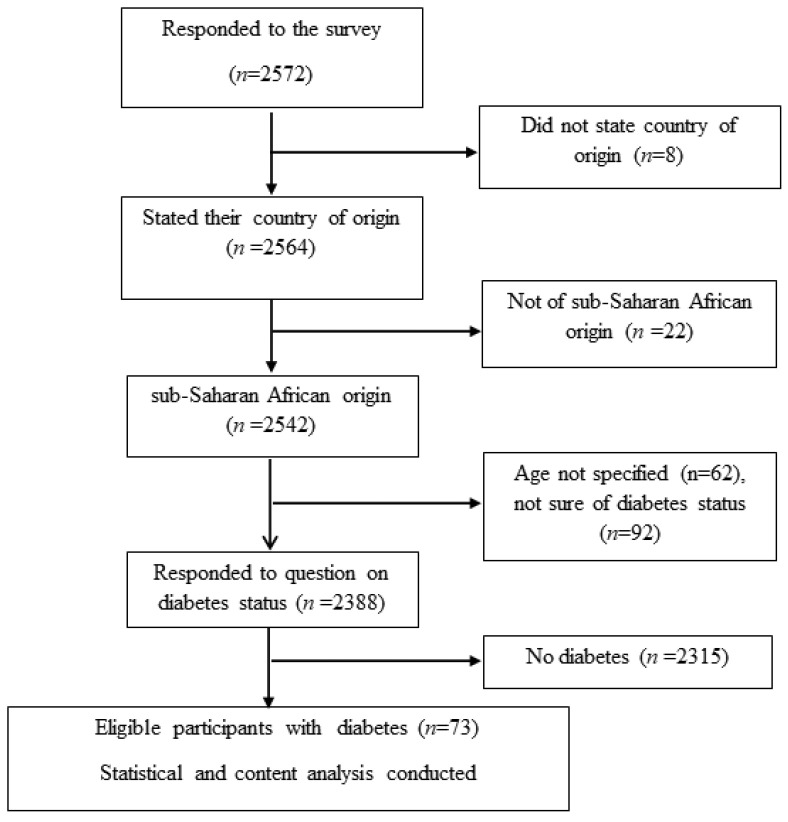
Flowchart of study participants.

**Figure 2 ijerph-19-07875-f002:**
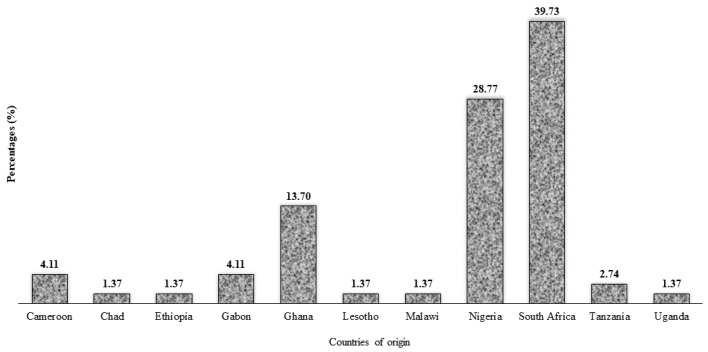
Distribution of study participants (*n* = 73) by country of origin.

**Figure 3 ijerph-19-07875-f003:**
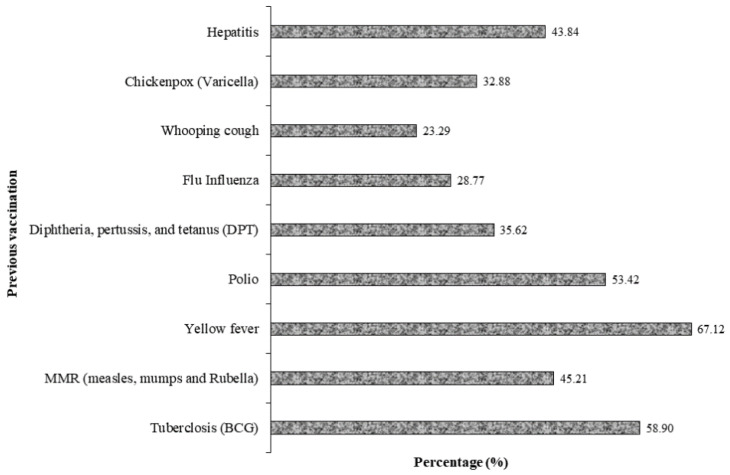
Percentage of previous vaccinations for other conditions among the study participants.

**Figure 4 ijerph-19-07875-f004:**
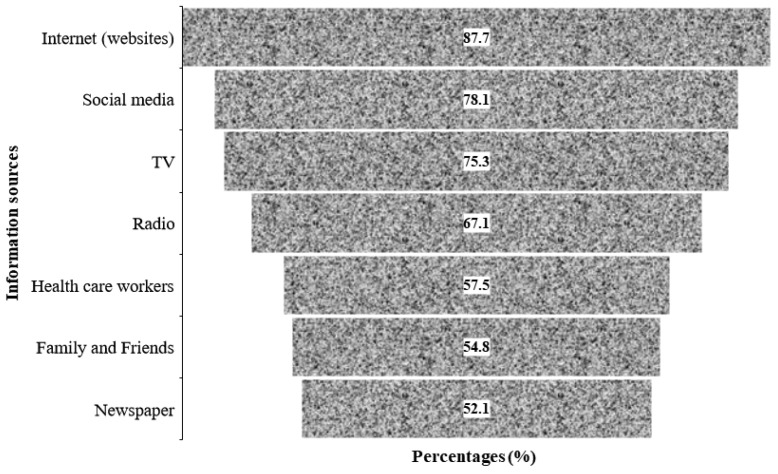
Main sources of COVID-19-related information used by the participants.

**Figure 5 ijerph-19-07875-f005:**
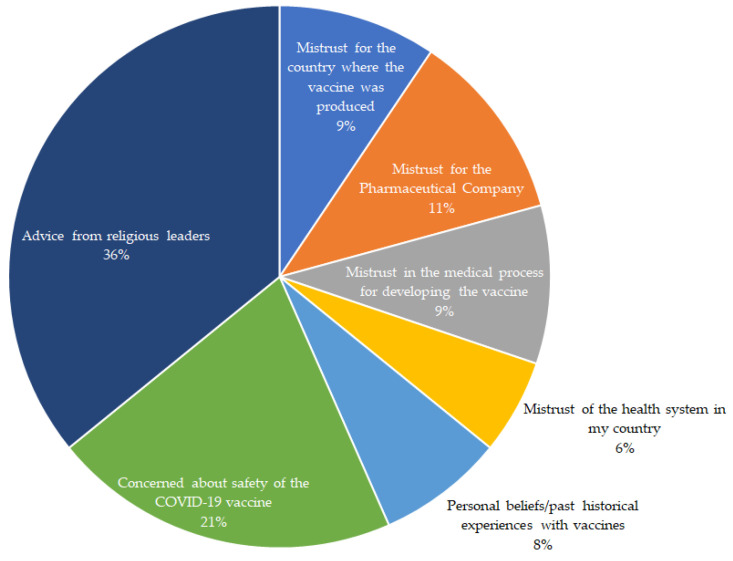
Main reasons for vaccine hesitancy or refusal among people with diabetes (*n* = 19).

**Table 1 ijerph-19-07875-t001:** Sociodemographic characteristics of study participants (*n* = 73).

Variables	Frequency *n* (%)
Demography	
**Age category in years**	
<35	10 (13.7)
≥35	63 (86.3)
**Sex**	
Males	48 (65.8)
Females	25 (34.2)
**Place of residence** ^ǂ^	
Local	62 (84.9)
Diaspora	9 (12.3)
**SSA region of origin**	
Central Africa	7 (9.6)
East Africa	5 (6.8)
Southern Africa	30 (41.1)
West Africa	31 (42.5)
**Marital status**	
Not married	20 (27.4)
Married/de facto	53 (72.6)
**Highest level of education**	
Secondary or less	6 (8.2)
University/diploma	35 (48.0)
Postgraduate (master’s/PhD)	32 (43.8)
**Employment status** ^ǂ^	
Unemployed	14 (19.2)
Employed	58 (79.5)
**Religion**	
Non-Christians	16 (21.9)
Christians	57 (78.1)
**Occupation** ^ǂ^	
Nonhealthcare sector	48 (65.8)
Healthcare sector	23 (31.5)
**Smoking status**	
Current smoker	9 (12.3)
Ex-smoker	11 (15.1)
Nonsmoker	53 (72.6)
**Previous vaccination**	
Yes	64 (87.7)
No/not sure	9 (12.3)

^ǂ^ frequencies do not add up to 100% due to some missing responses.

**Table 2 ijerph-19-07875-t002:** Other self-reported health conditions of study participants (*n* = 73).

Variables	Frequency *n* (%)
Any chronic disease ǃ	
Yes	46 (63.0)
No	27 (37.0)
Asthma ^ǂ^	
Yes	6 (8.2)
No	59 (80.8)
Hypertension ^ǂ^	
Yes	34 (46.6)
No	36 (49.3)
Sickle cell anemia ^ǂ^	
Yes	1 (1.4)
No	67 (91.7)
Obesity ^ǂ^	
Yes	15 (20.5)
No	55 (75.3)
Any heart condition ^ǂ^	
Yes	8 (11.0)
No	62 (85.0)
Kidney disease ^ǂ^	
Yes	2 (2.7)
No	66 (91.8)

ǃ = asthma, hypertension, obesity, kidney disease, sickle cell anemia, any heart condition; ^ǂ^ = few missing responses.

**Table 3 ijerph-19-07875-t003:** Awareness and risk perception of COVID-19 vaccine among study participants.

Variables	Frequency (%)
**Awareness of COVID-19 vaccination**	
Symptom of COVID-19	
Yes	47 (64.4)
No/not sure	26 (35.6)
Tested for COVID-19	
Yes	34 (46.6)
No	38 (52.1)
Tested positive for COVID-19 ^ǂ^	
Yes	14 (41.2)
No	20 (58.8)
Aware that COVID-19 vaccines have been developed	
Yes	71 (97.3)
No	1 (1.4)
**Have you been vaccinated against COVID-19**	
Yes	5 (6.8)
No	67 (91.8)
**Will you be willing to take COVID-19 vaccine when it becomes available in your country?**	
Yes (willing)	48 (65.8)
No/not sure (refusal/hesitancy)	19 (26.0)
**Risk perception of COVID-19 vaccination**	
Do you think COVID-19 virus is real	
Yes	68 (93.2)
No/not sure	5 (6.8)
COVID-19 vaccine can prevent COVID-19 infection and its complications	
Strongly agree	17 (23.3)
Agree	25 (34.2)
Don’t know	15 (20.5)
Disagree	8 (11.0)
Strongly disagree	1 (1.4)
Perception of risk of dying from COVID-19 infection	
Very high	13 (17.8)
High	25 (34.2)
Unlikely	7 (9.6)
Low	18 (24.7)
Very low	8 (11.0)
Perception of risk of becoming infected	
Very high	12 (16.4)
High	35 (47.9)
Unlikely	8 (11.0)
Low	16 (21.9)
Very low	2 (2.7)

^ǂ^ = denominators are those that were tested for COVID-19.

**Table 4 ijerph-19-07875-t004:** Association between COVID-19 vaccine hesitancy or refusal and the demographic variables of the participants. Only significant variables are shown.

Variables	No/Not Sure	Yes	*p*-Value
**Gender**			
Male	7 (14.6)	36 (75.0)	0.008
Female	12 (48.0)	12 (48.0)	
**SSA region of origin**			
Central Africa	2 (28.6)	5 (71.4)	0.045
East Africa	2 (40.0)	3 (60.0)	
Southern Africa	5 (16.7)	25 (83.3)	
West Africa	10 (32.3)	15 (48.4)	
**Place of residence**			
Diaspora	3 (33.3)	3 (33.3)	0.009
Local	15 (24.2)	44 (71.0)	
Have you been tested for COVID-19?	7 (20.6)	26 (76.5)	0.006
Are you concerned about the vaccine safety?	11 (64.7)	5 (29.4)	<0.001

Responses of those who answered ‘no’ or ‘not sure’ corresponding to ‘hesitancy and refusal’ to the question, ‘will you be willing to receive COVID-19 vaccine when it becomes available in your country?’ were merged. *p*-value are results of chi-squared analysis.

## Data Availability

The datasets analyzed during this study are available from the authors on reasonable request.

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
