# Peer review of "Analysis of Perception, Reasons, and Motivations for COVID-19 Vaccination in People with Diabetes across Sub-Saharan Africa: A Mixed-Method Approach"

_ijerph, 2022, doi:10.3390/ijerph19137875_

Round 1
Reviewer 1 Report
The authors have appropriately responded to the comments presented in the previous review. The manuscript is now acceptable for publication.
Author Response
Please see attachment

This manuscript is a resubmission of an earlier submission. The following is a list of the peer review reports and author responses from that submission.
Round 1
Reviewer 1 Report
This is a nicely written paper on an important topic. However, the authors have already published their results on vaccine hesitancy in African countries in excellent journals (eg. PLOS ONe: "Will Africans take COVID-19 Vacciantion" in December 2021). Now they are slicing their data to subgroups. The population of just 73 diabetics in several African countries is too small to any sound conclusions.
Reviewer 2 Report
This paper describes the results of a cross-sectional, online survey designed to determine hesitancy and unwillingness to COVID-19 vaccination among people with type 2 diabetes living in Sub-Saharan African countries.
While I think this is a very timely research topic, I have some concerns with this paper.
Major
1) 2.5
line 174-177
Please describe the METHOD in detail for the qualitative analysis part.
2) 3.3
Statistical Analysis
I recommend that the information presented in Table 4 be consulted by a statistician.
For example, if the number of persons in a cell is small, I believe that it is better to use Fisher's exact test.
I am not sure how a chi-square analysis could be performed for "Been tested for COVID-19" and "Concerned about the vaccine safety.
Furthermore, I could not understand which part of Table 4 corresponds to the contents of line 235-239.
3) 3.4
From Fig. 5, it seems that the main reason is "Advice from religious leaders," but why is this point not mentioned in either the results section or the discussion section?
4) 3.5
I would like to know why you made Fig. 6 a pyramid. On what basis is each factor placed in its respective position in the pyramid?
Minor
1) 2.4
Is the number of SSA countries 46 (line 140) or 26 (line 144)?
2) 3.2
I don't understand the meaning of line 214-216.
I think that the source from which they got the information is not the same as the source they rely on.
Regarding Fig. 4, internet (websites) can be public, such as the Ministry of Health, or private, such as social networking sites. What is the basis for the authors' conclusion that it is the Ministry of health?
Reviewer 3 Report
What are the limitations in regards to the sampling methods? How does using social media impact the generalizability of the study results?
In the statistical analysis section, the authors mention that "collinearity was assessed and no variable required exclusion due to weak association." It is not entirely clear if the authors mean that they assessed collinearity and found none (thus there was no need to exclude variables based on collinearity) AND they did not find any weak bivariate associations so they did not leave any of the originally intended variables out of the models or if they misunderstand assessing collinearity (or are just unclear on how they are presenting their understanding of collinearity).
In the results section, the first paragraph under the heading "3. Results" line 179-181 on p. 4, should be deleted. It looks to be a reminder of what should be included in a results section rather than an actual part of the manuscript.
Please left justify the words in the tables. It is very hard to read when they are centered.
A traditional pie chart would be much more compelling for figure 5.
It is very unclear how Figure 6 was developed. Is it supposed to be just a general feel for how the qualitative information was presented? Is there an actual relationship between the size of the row in the pyramid and the number of responses in that area based on qualitative assessment? It would be helpful if the authors could present the number or proportion of responses that were described within each layer of the pyramid.
Why did the authors not include a quantitative assessment of the relationship between having prior vaccinations and hesitancy for covid vaccintation? It appears to be quite relevant and should be included in the quantitative analysis. The authors describe this relationship in the discussion, but I do not see it in the table of results.
